# Evaluating the effectiveness of a brief digital procrastination intervention targeting university students in Sweden: study protocol for the Focus randomised controlled trial

Katarina Åsberg  , Marcus Bendtsen

**To cite:** Åsberg K, Bendtsen M. Evaluating the effectiveness of a brief digital procrastination intervention targeting university students in Sweden: study protocol for the Focus randomised controlled trial. *BMJ Open* 2023;**13**:e072506. doi:10.1136/bmjopen-2023-072506

Department of Health, Medicine and Caring Sciences, Linköping University, Linköping, Sweden

**Correspondence to**
Katarina Åsberg;
katarina.asberg@liu.se

## ABSTRACT

**Introduction** The concept of procrastination can be described as a conscious, yet irrational, postponement of important tasks or decisions—despite awareness that the delay may lead to negative consequences. Procrastination behaviours are common among university students and is often described as a failure of self-regulation, and the behaviour is associated with stress, symptoms of depression and anxiety, poorer academic performance and negative effects on overall health and well-being.

**Methods and analysis** A two-arm, parallel groups (1:1), single-blind randomised controlled trial will be conducted to assess the effectiveness of a brief digital procrastination intervention (Focus) among university students in Sweden. The intervention consists of a screening and feedback component based on Pure Procrastination Scale (PPS) score, allowing intervention participants to assess their current procrastination behaviours and receive behaviour change advice. Participants in the control group will be shown their total PPS score without any further feedback. Monte Carlo simulations (assuming a standardised effect of 0.35 Cohen's d of the intervention on the primary outcome, to at least 80% of the time estimate a posterior probability of effect of at least 95%) indicated that data from 1000 participants are required for analysis, meaning that 2000 participants are required to be randomised when assuming a 50% attrition rate. The primary outcome will be procrastination behaviour measured at 2 months postrandomisation. Secondary outcomes will be anxiety and stress symptoms and lifestyle behaviours. Outcomes will be analysed using multilevel regression models estimated using Bayesian inference.

**Ethics and dissemination** The study was approved by the Swedish Ethical Review Authority on 2022-08-24 (dnr 2022-00353). Students will be asked to give informed consent before participation and after having been given information about the study. The results will be submitted for publication in open access, peer-reviewed journals.

**Trial registration number** ISRCTN13533793

## INTRODUCTION

The concept of procrastination can be described as a conscious, yet irrational, postponement of important tasks or

## STRENGTHS AND LIMITATIONS OF THIS STUDY

⇒ Previous research on procrastination has mainly focused on examining mechanisms behind the behaviour, and to a lesser extent on evaluating support that could be helpful in changing procrastination behaviours.
⇒ The effectiveness of a novel brief digital intervention aiming to help college and university students change their procrastination behaviours will be evaluated.
⇒ The trial design necessitates self-assessment of behaviours (self-monitoring), which itself may influence behaviours, leaving a potential risk of bias since the control group will be offered a chance to self-assess their behaviours.
⇒ Attrition is a concern in digital intervention research and is anticipated to be a source of bias in the trial.

decisions—despite awareness that the delay may lead to negative consequences. To procrastinate has been defined by Steel as 'to voluntarily delay an intended course of action despite expecting to be worse off for the delay'.[1] For example, procrastinators tend to voluntarily keep busy with unrelated tasks instead of working to meet deadlines, leading to a short-term relief of undesired feelings associated with the prioritised task.

Procrastination is often described as a failure of self-regulation,[2] and the behaviour is associated with stress, symptoms of depression and anxiety,[3 4] poorer academic performance[5] and negative effect on overall health and well-being.[6] Previous research on procrastination has mainly focused on examining mechanisms behind the behaviour, for example, why people procrastinate and associations with personality traits such as impulsivity,[7] and to a lesser extent on evaluating support that could be helpful in changing procrastination behaviours.[8]

Although procrastination of health behaviours is relatively unexplored in the literature,[9] two examples of domain specific procrastination are 'bedtime procrastination' leading to insufficient sleep,[10] and 'exercise procrastination' associated with lower overall physical activity even after controlling for intentions and general procrastination.[11]

Procrastination behaviours are common among university students and previous studies indicate that approximately 50% of students are likely to procrastinate their academic assignments or tasks.[12 13] Also, in the general population, procrastination tends to be more common among younger adults aged 14–29 years.[4] For many students, the transition to higher education involves more academic demands and lesser external control of things such as scheduled hours versus attention competition from social and digital activities. Students are responsible for their own study technique, and previously used strategies such as 'last-minute-task performance' might create a breeding ground for procrastination, which in turn might end up in negative health consequences.

The majority of scientifically evaluated procrastination support has been based on cognitive–behavioural therapy (CBT) treatment programmes, delivered both in face to face and digital settings, and both with and without additional support from a therapist.[14–19] For example, a study by Rozental et al[14] evaluated guided and unguided self-help using internet based CBT (iCBT)-treatment in the general population and showed a moderate effect on procrastination. Another study from Rozental et al compared iCBT with group therapy among university students with severe procrastination symptoms, and showed no statistically significant differences between groups after treatment.[15] Eckert et al[16] evaluated a 2-week unguided online intervention for university students to overcome procrastination, with and without additional text messages, which showed small to medium effects on procrastination.

Since procrastination is common among university students, and since it may lead to ill health and have a negative impact on academic performance, there is a need to develop and evaluate a scalable, brief digital procrastination intervention to reach a general student population in need of support.

## Aims and objectives

The primary aim of this study, which is called Focus, is to estimate the effects of a brief digital procrastination intervention on procrastination behaviours among university students in Sweden. Secondary aims include estimation of the effects of the intervention on anxiety and stress symptoms, as well as lifestyle behaviours. The study objectives are to:

1. Estimate the effects of a brief digital procrastination intervention on procrastination behaviours.
2. Estimate the effects of a brief digital procrastination intervention on anxiety and stress symptoms.
3. Estimate the effects of a brief digital procrastination intervention on alcohol consumption, intake of candy and snacks, sugary drinks, and physical activity.
4. Investigate acceptability, perceived usefulness and general opinions of the intervention in terms of user experiences.

## METHODS AND ANALYSIS

A two-arm, parallel groups (1:1), single-blind randomised controlled trial (RCT) will be conducted to assess the effectiveness of a brief digital procrastination intervention targeting university students. This protocol includes relevant items from the Standard Protocol Items: Recommendations for Interventional Trials 2013 statement.[20]

### Study setting, recruitment and eligibility

The Focus study will be situated at colleges and universities in Sweden. Students at participating colleges and universities will be invited via email by their university student healthcare to participate in a study, in which they are asked to assess their procrastination. The email contains a hyperlink to a webpage presenting study information and informed consent materials (see online supplemental appendix A). Students who give informed consent will immediately be taken to a web-based questionnaire, which will collect baseline data and check for eligibility. Recruitment will begin in February 2023, and a final dataset will be available by the end of the semester of 2023.

#### Eligibility criteria

Students completing the baseline questionnaire and scoring 20 points or more on the Pure Procrastination Scale (PPS) will be eligible for randomisation into the trial. There are no validated (or recommended) cut-offs of the PPS that indicate which individuals have no procrastination issues, rather, it is a continuous measure of procrastination. However, we did not want to include individuals who scored very low (no procrastination) since they would not be the target group in a future intervention implementation. However, we did want to be inclusive and therefore set the cut-off relatively low at 20 points (note the range of Perceived Stress Scale (PSS) is 12–60), excluding only those who report none or very little procrastination. As the survey will be conducted in Swedish, students not familiar with the Swedish language will be implicitly excluded. Excluded students will be given access to the digital intervention but will not be randomised into the study and not followed-up. After completing the questionnaire, eligible participants will be randomised to one of the two conditions: the Focus intervention (intervention group) or a non-treatment condition (control group). Figure 1 outlines the study design.

### Interventions

#### Summary score and information (control group)

Participants allocated to the control group will be shown their total PPS score, with the minimum and maximum

Åsberg K, Bendtsen M. *BMJ Open* 2023;**13**:e072506. doi:10.1136/bmjopen-2023-072506

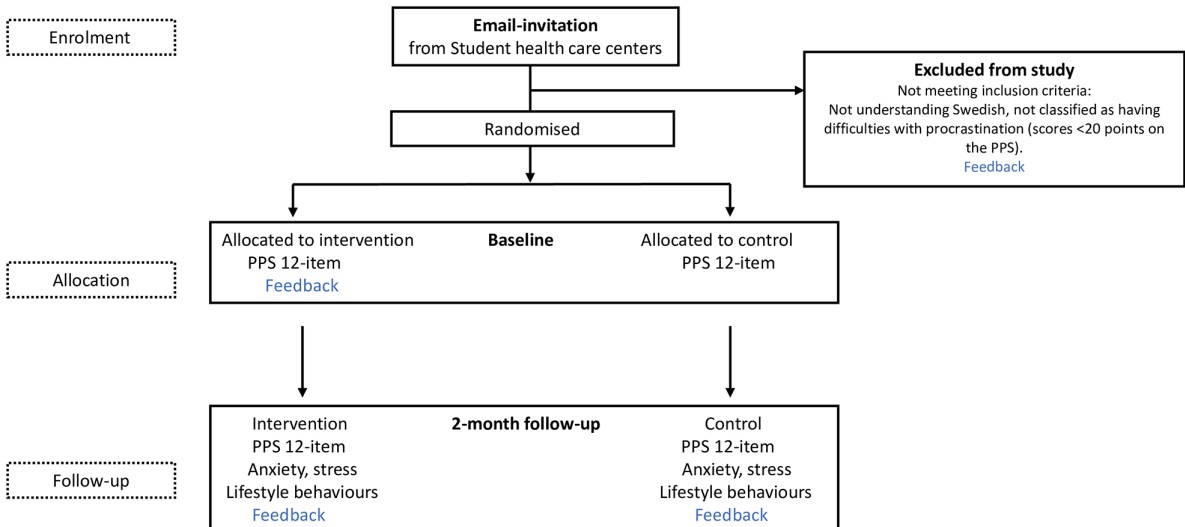

**Figure 1** Study design flow chart. PPS, Pure Procrastination Scale.

on the scale presented, and a recommendation to read more about procrastination at their local student health-care website.

### The Focus intervention (intervention group)

Participants allocated to the intervention group will be given the same information as the control group, but also be given access to the novel Focus intervention.

The Focus intervention is a digital procrastination intervention which is delivered online to students email and consists of a screening and feedback component, allowing participants to assess their current procrastination behaviours and receive behaviour change advice. The intervention is designed with inspiration from both previous research on procrastination treatment programmes using internet-based CBT (iCBT),[14 15] and on the concept of brief alcohol interventions, which have been widely researched among university students in Sweden.[21]

The intervention was designed in iterative steps following the first stage ('understanding the behaviour') of the Behaviour Change Wheel (BCW) as a theoretical framework.[22 23] Procrastination behaviour was understood by defining the problem in behavioural terms, specifying the target behaviour and identifying what needs to change in dialogue with professionals at the student healthcare centres during several multidisciplinary group meetings. Intervention content was also designed and developed in collaboration with the student healthcare centres.

The intervention starts with participants assessing their procrastination behaviour using the PPS questionnaire. Following the assessment, participants are given feedback on their current procrastination behaviours and personalised behaviour change advice. The items of the PPS have been found to load onto a 3-factor model[24] where items 1–3 relate to decisional delay, items 4–8 to implemental delay and items 9–12 to timeliness and promptness delay. The feedback and advice given to participants as part of the digital intervention is therefore tailored in accordance

with these latent constructs, allowing for prioritisation of advice based on the highest scoring construct.

The personalised advice component is based on the self-assessment and consists of graphical, written and interactive content. The feedback aims to enhance self-reflection and self-evaluation and aims to stimulate self-reinforcement on participants procrastination behaviour, in line with self-regulation theory.[25] Figure 2 visualises screenshots of the intervention.

### Outcomes

The following outcome measures will be used. All study questionnaires can be found in online supplemental appendix B.

### Primary outcome measures
► Procrastination behaviour.

### Secondary outcome measures
► Anxiety.
► Stress.
► Weekly alcohol consumption.
► Monthly frequency of heavy episodic drinking.
► Weekly consumption of candy and snacks.
► Weekly consumption of sugary drinks.
► Weekly moderate to vigorous physical activity (MVPA).

### User evaluation
► Perceived acceptability and usefulness of the intervention among users.

### Primary and secondary outcome measures

Procrastination behaviour will be assessed by asking participants to complete the PPS 12-item PPS. Each item is scored on a five-point Likert scale with a higher score indicating greater difficulties with procrastination (12–60). The scale was originally developed by Steel[26] and has been translated to Swedish and validated by Rozental et al,[27] showing good internal consistency (Cronbach's alpha=0.78).

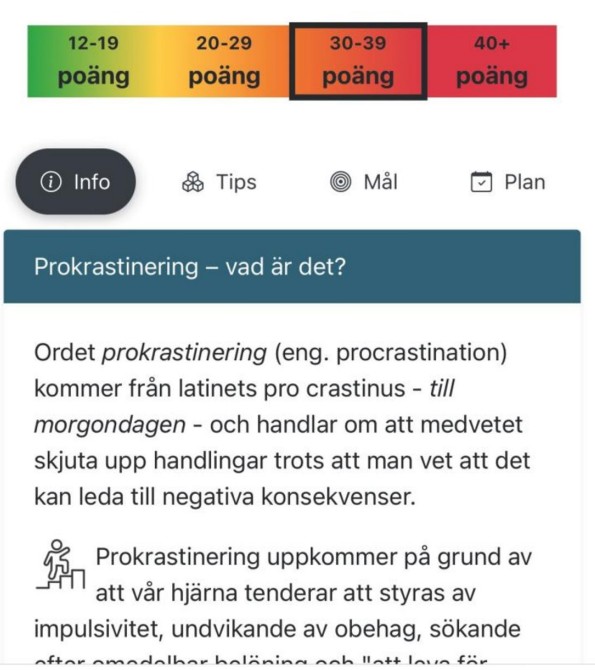

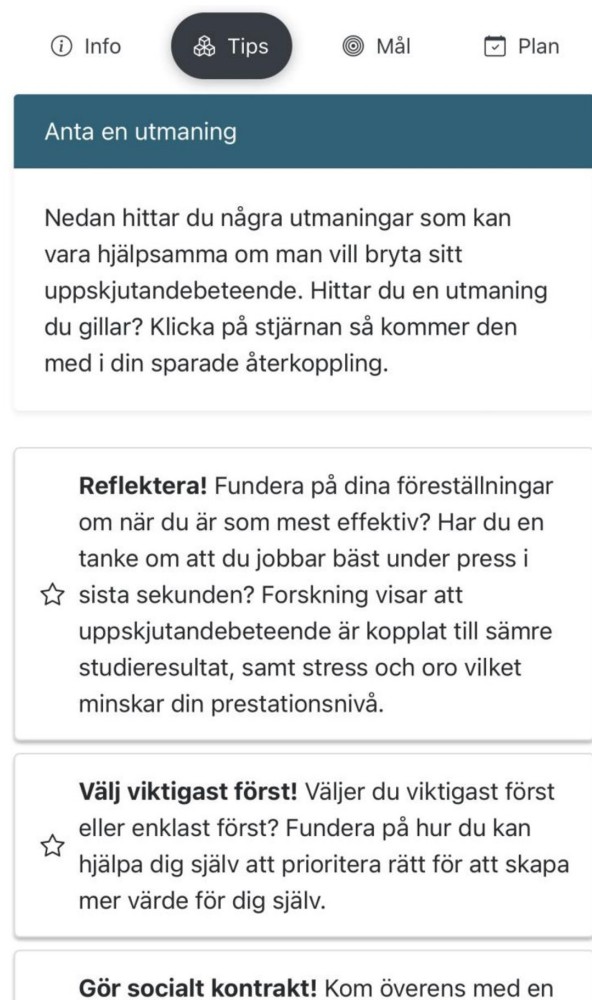

**Figure 2** Screenshots of the intervention.

Anxiety will be assessed using the validated seven item Generalised Anxiety Disorder scale,[28] asking participants how often they, over the last 2 weeks, have been bothered by seven different anxiety related problems. Perceived stress will be assessed using the four item short form PSS, designed to measure experienced levels of stress,[29] and considered a useful instrument for assessing stress perception levels in the general population.[30]

Procrastination behaviours are potentially a barrier for healthy lifestyle behaviours; thus we have added lifestyle behaviour outcome measures to explore this hypothesis. Weekly alcohol consumption will be assessed by asking participants the number of standard drinks of alcohol they consumed last week (short-term recall method). Frequency of heavy episodic drinking will be assessed by asking participants how many times they have consumed four or more standard drinks of alcohol on one occasion the past month. These two outcomes are both part of the proposed core outcome set for brief alcohol interventions.[31]

Diet and physical activity will be measured utilising a questionnaire based on the previously published questionnaire designed by the National Board of Health and Welfare in Sweden, and was further modified to also include portion sizes.[32] Candy and snacks will be measured using a single question regarding number of servings consumed last week, and sugary drinks consumption will be measured by a question regarding the number of units (33 cl) of sugary drinks participants consumed the past week. Moderate and vigorous physical activity will be estimated by summing responses to two questions regarding the number of minutes spent on moderate and vigorous physical activity, respectively, during the past week.

At 2 months follow-up, participants allocated to the intervention group will be asked three questions about perceived acceptability and usefulness of the intervention.

### Participant timeline and follow-ups

Recruitment, informed consent, eligibility screening, randomisation, baseline assessment, intervention and follow-ups will all be automated and computerised. Two email reminders will be sent 1 week apart. Follow-up at 2 months after randomisation will be initiated by sending an email with a hyperlink to a web-based questionnaire. Non-responders to follow-up will receive up to three follow-up email reminders. A study timeline is presented

Åsberg K, Bendtsen M. *BMJ Open* 2023;**13**:e072506. doi:10.1136/bmjopen-2023-072506

| | STUDY PERIOD | | | |
|---|---|---|---|---|
| | Enrolment | Allocation | Post-allocation | Close-out |
| **TIMEPOINT** | **0** | **0** | **0** | **2 months** |
| **ENROLMENT:** | | | | |
| Informed consent | X | | | |
| Eligibility screen | X | | | |
| Allocation | | X | | |
| **INTERVENTIONS:** | | | | |
| *Digital intervention* | | X ← | ——————→ | |
| *Control condition* | | X ← | ——————→ | |
| **ASSESSMENTS:** | | | | |
| *Baseline questionnaire* | X | | | |
| *Pure procrastination scale* | X | | | X |
| *Lifestyle questionnaire* | | | | X |
| *User evaluation questionnaire* | | | | X |

**Figure 3** SPIRIT figure presenting participant timeline. SPIRIT, Standard Protocol Items: Recommendations for Interventional Trials.

in figure 3. Note that this is the first ever effectiveness study of a brief digital intervention for procrastination of its kind, and as such we have no previous studies to guide us on how long intervention effects may persist. We took a pragmatic approach and decided to keep the follow-up interval short enough to stay within the same academic semester, but still enough removed from the intervention itself to decide if there were any lasting effects.

## Assignment of interventions

Participants will be randomised using block randomisation (with random block sizes of 2 and 4). The randomisation sequence will be computer generated, and neither research personnel nor participants will be able to manipulate the sequence. As all study procedures are automated, thus randomisation cannot be subverted.

## Sample size

The data collected in this trial will be analysed using Bayesian analysis,[33] which delivers a posterior probability of effect regardless of the number of participants included. We will not use null hypothesis testing; therefore, a traditional power calculation is not required to determine the required sample size to control type I and II errors. Instead, we conducted a Monte Carlo simulation study, assuming a standardised effect of 0.35 Cohen's d of the intervention on the primary outcome (procrastination behaviour). To at least 80% of the time estimate a posterior probability of effect of at least 95%, the Monte Carlo simulations indicated that data from 1000 participants are required for analysis, meaning that 2000 participants are required to be randomised when assuming a 50% attrition rate. Note that this intervention and trial is novel, and as such there are no previous studies to rely on the determination of effect size. However, a previous study comparing guided and unguided self-help

using iCBT with a wait-list control observed effects sizes ranging from 0.5 to 0.7 Cohen's d.[14] We anticipate that our brief intervention will be less effective than a full therapy programme, and we have thus assumed a small to medium Cohen's d of 0.35.

## Data analysis

All analyses will be conducted keeping participants in the groups to which they were randomised (intention to treat). We will conduct available data analyses and use multiple imputation with chained equations (with 200 imputed data sets using 30 iterations) to impute missing data for imputed analyses. Both available data and imputed analyses will be used to interpret findings. All models will be estimated using Bayesian inference, with the posterior probability of effect being presented along with posterior medians and 95% compatibility intervals. All models will include an adaptive intercept for each university (multilevel), with standard normal priors.

The primary outcome (PPS) will be standardised and analysed using linear regression, adjusted for age, gender and PPS at baseline. Anxiety, stress, weekly consumption of candy and snacks, and MVPA will be modelled equivalently as PPS. Frequency of heavy episodic drinking, weekly alcohol consumption and weekly consumption of sugary drinks will be analysed using negative binomial regression, adjusted for age, gender and PPS at baseline. Student-t priors will be used for covariate coefficients, normal priors will be used for adaptive intercepts with half-student-t hyperpriors for the SD, and half-student-t priors will also be used for main SDs and dispersion parameters. All student-t priors will have 3 df and a scale of 2.5.

Interaction analyses will be conducted between treatment allocation and age, gender and baseline PPS. Models will be compared using the widely applicable information criterion to decide if interaction models are informative in comparison to non-interaction models.

Attrition analyses will be conducted to investigate systematic differences between responders and non-responders. Logistic regression will be used to estimate the OR of not responding to follow-up with respect to age, gender and baseline PPS. The same priors as the primary analyses will be used.

## Patient and public involvement

Previous research, interviewing university students about their experiences of health and health-related behaviours, highlighted procrastination and study-related stress as a common barrier for not engaging in healthy routines such as physical activity and healthy cooking.[34] This study was designed and developed in collaboration with professionals at student healthcare centres across Sweden. During the process of designing the intervention content and usability testing of the digital support, approximately 10 professionals took part in multidisciplinary group meetings during 2022. These meetings defined procrastination in behavioural terms, specified the target

behaviour, identified what needs to change, and ways to achieve this using a digital tool.

## ETHICS AND DISSEMINATION

The study was approved by the Swedish Ethical Review Authority on 2022-08-24 (dnr 2022-00353). Students will be asked to give informed consent before participation and after having been given information about the study. The results will be submitted for publication in open-access, peer-reviewed journals.

## DISCUSSION

This RCT aims to estimate the effects of a brief digital procrastination intervention on procrastination behaviours among university students in Sweden. It will add findings on how to support students to overcome procrastination—something that is inadequately evaluated. This is so despite procrastination behaviours being common among university students, and known to cause ill health, distress and negatively impact academic performance.[3 5] Digital interventions could fulfil these needs and are scalable to a large group of individuals at a low cost. The low cost of delivering digital interventions,[35] and the low participant burden of completing a brief screening questionnaire, justifies even small effects from the intervention.

### Generalisability

This study is carried out in a college and university setting, reaching students via communication channels used in their academic and daily lives. Students signing up to participate in the study do not represent a population consisting entirely of individuals actively seeking help, and the inclusion criteria are relatively broad for inclusion. The study can, therefore, be understood as an effectiveness trial, which aims to estimate the effects of the intervention in a naturalistic setting. It should be noted that the intervention materials are designed for college and university students, and therefore, generalisability outside the target population is not anticipated. Also, being a student in Sweden is not necessarily comparable to being a student in other countries, and so generalisability to other countries may be limited.

### Limitations

The trial design necessitates self-assessment of procrastination behaviours among all participants prior to randomisation. Previous research has shown that assessment of behaviours (self-monitoring) may have an effect on self-reported behaviour,[36 37] thus there is risk of diluting effects by the control group being offered a chance to self-assess their procrastination behaviours. This is, however, unavoidable in order to maintain the naturalistic design of the study and to conduct data collection. Most conservatively, estimated effects may therefore be viewed as caused by the feedback and advice component only, assuming no interaction with self-assessment.

Attrition bias is a concern in digital intervention research, including this study. We are anticipating high attrition at 50%, which is difficult to recover from even using modern methods of imputation. Attrition analyses will reveal if there is evidence of systematic differences among responders and non-responders; however, the study will nevertheless be limited by the risk of attrition bias.

Finally, we have chosen a pragmatic follow-up interval of 2 months. This means that we will not be able to identify earlier effects of the intervention which may have waned.

### Public health relevance

Procrastinating affects both physical and mental health negatively and can, apart from increased stress, involve risk behaviours, such as excessive alcohol consumption or tobacco use as stress-relieving strategies. Procrastination behaviour might also out-compete good health habits, such as sleep hygiene, regular physical activity or healthy eating habits, due to high level of stress. By quitting postponing important commitments students increase the chance of succeeding with their studies, decreasing their risk of mental illness, exhaustion and anxiety-related problems. Today, there is a lack of support to help students take control of their procrastination behaviours. Therefor, and due to the high prevalence of procrastination among students, it is valuable to evaluate a scalable digital procrastination support for university students.

**Acknowledgements** We would like to acknowledge the student health care professionals at the participating universities who provided us with their time and shared their experiences.

**Contributors** Study objectives, outcomes, trial design and analysis plan were decided by MB and KÅ, as well as development and conceptualisation of intervention materials. KÅ drafted the protocol, which was revised by MB. Both authors contributed with intellectual content and approved the final version. KÅ and MB will be responsible for data collection and statistical analysis.

**Funding** This trial is conducted under the auspices of the Swedish Research Council for Health, Working Life, and Welfare (Grant number 2018-01410).

**Disclaimer** The funders had no role in study design, data collection and analysis, decision to publish, or preparation of the manuscript.

**Competing interests** MB owns a private company (Alexit AB) that maintains and distributes evidence-based lifestyle interventions to be used by the public and in health care settings. Alexit AB played no role in developing the intervention, study design, data analysis, data interpretation or writing of this report.

**Patient and public involvement** Patients and/or the public were involved in the design, or conduct, or reporting, or dissemination plans of this research. Refer to the Methods section for further details.

**Patient consent for publication** Not applicable.

**Provenance and peer review** Not commissioned; externally peer reviewed.

**ORCID iDs**

Katarina Åsberg http://orcid.org/0000-0003-2194-6789

Marcus Bendtsen http://orcid.org/0000-0002-8678-1164

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
