## [Reviewer comments · BMJ Open]

ARTICLE DETAILS

TITLE (PROVISIONAL)	Evaluating the effectiveness of a brief digital procrastination intervention targeting university students in Sweden: study protocol for the Focus randomised controlled trial
AUTHORS	Åsberg, Katarina; Bendtsen, Marcus

VERSION 1 – REVIEW

REVIEWER	Wu, Hongbin Peking University
REVIEW RETURNED	23-Apr-2023

GENERAL COMMENTS	Thank you for providing the protocol entitled "Evaluating the effectiveness of a brief digital procrastination intervention targeting university students in Sweden: study protocol for the Focus randomized controlled trial". I would like to offer the following feedback for your consideration. This protocol describes a randomized controlled trial of a digital procrastination intervention targeting university students in Sweden. The protocol aims to evaluate the effects of the intervention on procrastination behavior and other related outcomes such as anxiety, stress symptoms, and lifestyle behaviors. The protocol was approved by the Swedish Ethical Review Authority. However, there are some areas in which the study could be improved. First, as noted by the authors in the Limitations section, the study relies on self-reported measures which may dilute the effects as the control group being offered a chance to self-assess their procrastination behaviors. Additionally, attrition is a concern, as high procrastinators who failed to participate in the follow-up may have resulted in an overestimation of the treatment effects. Second, the study mentions that students scoring 20 points or more on the Pure Procrastination Scale (PPS) will be eligible for randomization into the trial. It is necessary to provide an explanation for the selection of this criterion as it directly affects the group entering the experiment and subsequently influences the treatment effects. Third, the study conducted follow-up on participants for 2 months after randomization. The reason for selecting a 2-month timeframe should be explained. One possibility is that treatment effects may have been demonstrated earlier and disappeared in the second month.
--

	Fourth, the study assumes a standardized effect of 0.35 Cohen's d of the intervention on the primary outcome (procrastination behavior) and determines that 1000 participants are required for analysis. However, it is unclear how the effect size of 0.35 Cohen's d was determined. The authors should provide an explanation or justification for this selection. Fifth, the study does not provide specific information on the content or format of the digital intervention, which may limit the ability to replicate or modify the intervention in other settings. Finally, the study is limited to Swedish university students, which may restrict the generalizability of the research findings to other populations or settings. Overall, the manuscript provides a comprehensive study protocol for evaluating the effectiveness of a digital procrastination intervention. However, the concerns raised above need to be addressed to improve the scientific rigor and impact of the study.
--	--

REVIEWER	ZHOU, YING Beijing Open University
REVIEW RETURNED	06-May-2023

GENERAL COMMENTS	 1. Thank you for the hard work in designing and carrying out this interesting study. 2. Why do you want to involve lifestyle measures in the outcomes such as alcohol consumption, intake of candy and snacks? 3. After the post assessment 2 months later, should there be a third assessment as follow-up to make sure the intervention effects would be kept in the long run?
--

VERSION 1 – AUTHOR RESPONSE

Reviewer #1:

Thank you for providing the protocol entitled "Evaluating the effectiveness of a brief digital procrastination intervention targeting university students in Sweden: study protocol for the Focus randomized controlled trial". I would like to offer the following feedback for your consideration. This protocol describes a randomized controlled trial of a digital procrastination intervention targeting university students in Sweden. The protocol aims to evaluate the effects of the intervention on procrastination behavior and other related outcomes such as anxiety, stress symptoms, and lifestyle behaviors. The protocol was approved by the Swedish Ethical Review Authority. However, there are some areas in which the study could be improved.

Q1. First, as noted by the authors in the Limitations section, the study relies on self-reported measures which may dilute the effects as the control group being offered a chance to self-assess their procrastination behaviors. Additionally, attrition is a concern, as high procrastinators who failed to participate in the follow-up may have resulted in an overestimation of the treatment effects.

A1. We agree that these are limitations of the study and, as the reviewer points out, we have discussed them in the limitations section of the protocol. Although there may be some proxy measures of procrastination that could be considered objective, e.g. study results, the PPS is a validated measure that captures more than just a single consequence of procrastination. Naturally, our findings will be reported in light of these limitations.

Q2. Second, the study mentions that students scoring 20 points or more on the Pure Procrastination Scale (PPS) will be eligible for randomization into the trial. It is necessary to provide an explanation for the selection of this criterion as it directly affects the group entering the experiment and subsequently influences the treatment effects.

A2. There are no validated (or recommended) cut-offs of the PPS that indicate which individuals have no procrastination issues, rather, it is a continuous measure of procrastination. However, we did not want to include individuals who scored very low (no procrastination) since they would not be the target group in a future intervention implementation. However, we did want to be inclusive and therefore set the cut-off relatively low at 20 points (note the range of PSS is 12-60), excluding only those who report none or very little procrastination. In other studies, a cut-off of 40 has been used, meaning that only those who report having quite severe procrastination were included, however, we did want to avoid this stringent inclusion criteria here. We have added our rationale to the manuscript in the 'Eligibility criteria' section, please see page 4.

Q3. Third, the study conducted follow-up on participants for 2 months after randomization. The reason for selecting a 2-month timeframe should be explained. One possibility is that treatment effects may have been demonstrated earlier and disappeared in the second month.

A3. This is the first ever effectiveness study of a brief digital intervention for procrastination of its kind, and as such we have no previous studies to guide us on how long intervention effects may persist. We took a pragmatic approach and decided to keep the follow-up interval short enough to stay within the same academic semester, but still enough removed from the intervention itself to decide if there were any lasting effects. We have added this rationale to the 'Participant timeline and follow-ups' section. Further, we agree with the reviewer that treatment effects may have been demonstrated earlier and have added this limitation to the 'Limitation' section, please see page 10.

Q4. Fourth, the study assumes a standardized effect of 0.35 Cohen's d of the intervention on the primary outcome (procrastination behavior) and determines that 1000 participants are required for analysis. However, it is unclear how the effect size of 0.35 Cohen's d was determined. The authors should provide an explanation or justification for this selection.

A4. As we pointed out in response A3 above, this is the first full scale randomised controlled trial of a brief digital procrastination intervention of its kind, and so we have no previous studies to guide this decision. In a study comparing guided and unguided self-help using Internet-based cognitive behaviour therapy with a wait-list control, effects sizes ranging from 0.5 to 0.7 of Cohen's d were observed [1]. We anticipate that our brief intervention will be less effective than a full ICB therapy program, and thus assumed a small to medium Cohen's d of 0.35. We have added this information to the manuscript in the 'Sample size' section, please see page 7.

Q5. Fifth, the study does not provide specific information on the content or format of the digital intervention, which may limit the ability to replicate or modify the intervention in other settings.

A5. We have added a clarifying sentence regarding the intervention in section 'Interventions', please see page 5. There we now describe the intervention as being online and delivered via email. In the same section we describe the intervention with screenshots. Intervention materials are all in Swedish however if needed we could have these translated and added as supplementary materials. We ask the editor to please advise.

Q6. Finally, the study is limited to Swedish university students, which may restrict the generalizability of the research findings to other populations or settings.

A6. The intervention is designed for college and university students, and the content is focused on study technique etc. Thus, we do not anticipate that the intervention should generalise beyond this target population. The trial design is pragmatic and external validity is therefore stronger than in more controlled efficacy trials. Naturally, the context in which Swedish students live and study may be different from other countries, and so we have mentioned this in the 'Generalisability' section, please see page 10.

Reviewer #2:

Q1. Thank you for the hard work in designing and carrying out this interesting study.

A1. Thank you.

Q2. Why you wanna involve the lifestyle measures in the outcomes such as alcohol consumption, intake of candy and snacks?

A2. Procrastination behaviours are potentially a barrier for healthy lifestyle behaviours; thus we have added lifestyle behaviour outcome measures to explore this hypothesis. This statement is added to the 'Primary and secondary outcome measures' section, please see page 6, to clarify the involvement of lifestyle behaviour outcome measures.

Q3. After the post assessment 2 month later, should there be a third assessment as follow-up to make sure the intervention effects would be kept in the long run?

A3. While it would be interesting to study the longer term effects, we decided against having a second follow-up mainly for two reasons. First, we anticipate that attrition will be even lower in a second follow-up and would make valid inference about effects difficult. Second, it would mean having a measure of the interventions effect after having exposed the control group to assessment twice, which would, due to assessment reactivity, dilute the effects even further.

References

1. Rozental A, Forsell E, Svensson A, Andersson G, Carlbring P. Internet-based cognitive-behavior therapy for procrastination: A randomized controlled trial. J Consult Clin Psychol. 2015 Aug;83(4):808–24.

VERSION 2 – REVIEW

REVIEWER	Wu, Hongbin Peking University
REVIEW RETURNED	09-Jul-2023
GENERAL COMMENTS	Thank you for addressing my concerns.